# HALO: Hadamard-Assisted Lower-Precision Optimization for LLMs

**Saleh Ashkboos**[*][†]
ETH Zurich

**Mahdi Nikdan**[*]
ISTAustria

**Soroush Tabesh**[*]
ISTAustria

**Roberto L. Castro**
ISTAustria

**Torsten Hoefler**
ETH Zurich

**Dan Alistarh**
ISTAustria & Neural Magic

## Abstract

Quantized training of Large Language Models (LLMs) remains an open challenge, as maintaining accuracy while performing all matrix multiplications in low precision has proven difficult. This is particularly the case when *fine-tuning pre-trained models*, which can have large weight, activation, and error (output gradient) outlier values that make lower-precision optimization difficult. To address this, we present HALO, a new quantization-aware training approach for Transformers that enables accurate and efficient low-precision training by combining 1) strategic placement of Hadamard rotations in both forward and backward passes, which mitigate outliers, 2) high-performance kernel support, and 3) FSDP integration for low-precision communication. Our approach ensures that all large matrix multiplications during the forward and backward passes are executed in lower precision. Applied to LLAMA-family models, HALO achieves near-full-precision-equivalent results during fine-tuning on various tasks, while delivering up to $1.41\times$ end-to-end speedup for full fine-tuning on RTX 4090 GPUs. HALO efficiently supports both standard and parameter-efficient fine-tuning (PEFT). Our results demonstrate the first practical approach to fully quantized LLM fine-tuning that maintains accuracy in INT8 and FP6 precision, while delivering performance benefits.

## 1 Introduction

The high performance of large language models (LLMs) across a wide series of tasks comes with considerable computational costs; reducing them is one of the key directions in Machine Learning Systems research [9; 16; 21]. For *LLM inference*, a standard acceleration approach has been the *quantization* of weights and activations, which reduces the precision at which they are stored and potentially also computed over, e.g. [17; 42; 1; 35; 3].

By contrast, much less is known about quantization in the context of LLM *training*. While the recent DeepSeek breakthrough [11] has showed that 8-bit floating-point (FP8) pre-training of large models can be done both accurately and efficiently, it is not known how to extend this result using the more widely-supported 8-bit integer (INT8) format, or lower-precision 6-bit formats such as MXFP6 [38; 41]. Intuitively, accurate and efficient quantized training is much more challenging than quantized inference: while in inference a single large matrix multiplication occurs in lower precision, for training, all three matrix multiplications (one during the forward pass and two during back-propagation) must be executed in lower precision. This creates critical issues both in terms of *accuracy*—as quantizing weights, activations, and errors can induce significant training instability—

---

[*]Equal contribution
[†]Correspondence to: saleh.ashkboos@inf.ethz.ch

39th Conference on Neural Information Processing Systems (NeurIPS 2025).

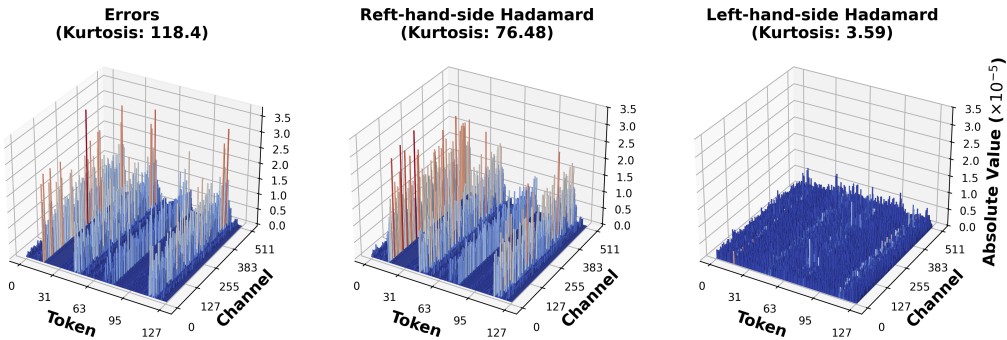

Figure 1: Largest magnitudes of first 512 channels in output gradients, or errors, of the mlp output (`down_proj`) in the 15-th layer of LLAMA3-8B model over 128 tokens in the 60th step of the ViGGO fine tuning (see Appendix A.4 for other data types). The outliers are propagated across columns and can be mitigated after applying left-hand-side Hadamard transformations.

but also in terms of *performance*—as the overheads of switching between representations can negate the performance gains of lower-precision computation.

This accuracy problem is especially challenging in the popular practical scenario where the user *fine-tunes a pretrained model*: as we illustrate in this paper, since pretrained LLMs often have large outlier values in the weight, activation, and error distributions [37; 42; 34; 28], stable results are much harder to achieve compared to from-scratch quantized pre-training.

**Contributions.** We present a new quantized training technique called Hadamard-Assisted Low-precision Optimization (HALO), which modifies the structure of Transformer-based models [36] in transparent fashion, allowing them to be fine-tuned in lower precision (INT8, or even FP6), with minimal accuracy loss. Importantly, we do so while performing *all large matrix multiplications* in lower precision. HALO applies equally well to full and parameter-efficient fine-tuning.

HALO starts from an in-depth analysis of the quantization error sensitivity of different internal representations (weight, input, and output gradients) during back-propagation [33; 22]. We identify the forward pass and output gradients as key sources of sensitivity to quantization, each exhibiting distinct outlier patterns. Specifically, while applying a Hadamard transformation is sufficient to mitigate outliers in the forward pass, addressing outliers in the output gradients during the backward pass requires applying the Hadamard matrix from the left side, referred as *Left-hand-side Hadamard* (Figure 1). This results in two "levels" of HALO, which can be chosen based on precision, data format, and acceptable accuracy drop. We complement our algorithmic contributions with efficient kernel support, allowing for computational speedups. In addition, we integrate HALO with the Fully Sharded Data Parallel (FSDP) scheme, to enable further savings via low-precision communication.

In summary, our contributions are as follows:

1. We consider the challenging problem of quantized fine-tuning of a pre-trained model using only lower-precision multiplications, and analyze the impact of quantization errors on model accuracy during training, specifically linking them to the presence of outliers across various model dimensions and internal states. Starting from this analysis, we propose using left- and right-hand-side Hadamard transforms to address outliers over matrix rows and columns, respectively. We show how these transforms can be inserted "strategically" in the Transformer architecture for both forward and backward passes, minimizing overheads.

2. We then explore different *outlier protection levels* for HALO, based on the number and placement of Hadamard transforms during training: Intuitively, HALO-1 strikes a trade-off between accuracy and performance that is well-suited for distributions with moderate dynamic ranges (such as FP6). In contrast, HALO-2 employs rotations to protect *all multiplications* during both the forward and backward passes, making it more suitable for integer representations (such as INT8). Importantly, the levels are independent of the precision used, allowing us to adjust them to recover final accuracy while maximizing end-to-end speedup. HALO is compatible with *full fine-tuning (FFT)* and *PEFT* methods, and is backed by new efficient GPU kernel implementations, currently aimed at NVIDIA RTX GPUs. Moreover, the fact that all computation happens in quantized form offers

the opportunity to also perform quantized communication during sharded (FSDP) training, for which we also add support in HALO.

3. We examine the accuracy and performance of HALO for fine-tuning LLAMA and Qwen-family models [13; 43], via both FFT and PEFT. We observe that HALO closely tracks the accuracy of full-precision variants across a wide series of tasks, improving upon the best known prior methods [38; 41] on the more challenging INT8 and FP6 formats. We provide performance measurements per module and end-to-end, with peak speedups of $1.82\times$ and $1.41\times$ for INT8, relative to a well-optimized half-precision baseline.

Overall, our results show for the first time that it is possible to perform fast and accurate fine-tuning while the majority of the forward-backward computation (all linear modules) is in lower precision, even if the model initially has an outlier structure in weights, activations, and errors.

## 2 Background

**Related Work.** Inference quantization methods aim to compress either the model weights [17; 14; 35] or jointly quantizing weights and activations, allowing for lower-precision forward computation [42]. It is known that activation quantization is challenging due to "outlier features" [37] of much larger magnitude. Recent works apply transforms to mitigate such features on top of pre-trained models. Chee et al. [5] observed that the magnitude of the largest elements (*outliers*, roughly defined as values $\geq 10$ larger than the value average in the considered tensor) in the weight matrices $\mathbf{W}$ that we wish to quantize for inference can be reduced by applying orthogonal transformations to $\mathbf{W}$ from both sides, a method known as *Incoherence Processing* [6]. QuaRot [3] mitigates the impact of outliers in both weights and activations by applying (randomized) Hadamard rotations that are fused into the weights—a technique known as *computational invariance*. This approach enables most inference computations to be performed in 4 bits without incurring the transform overhead during inference. SpinQuant [23] builds on a similar idea but *trains* a subset of orthogonal transformations. Unfortunately, *the QuaRot approach cannot be used for low-precision training* (see Section 3.1).

Performing *low-precision computation during the backward pass* is strictly harder than quantized inference, for instance due to the very high dynamic range of gradients with respect to layer outputs [24; 7]. LM-FP8 [32] trains large-scale models from scratch using FP8 on 40-100B of tokens, while Fishman et al. [15] demonstrates that FP8 training becomes unstable when training with over >250B tokens, and proposes a new activation function to address this issue. As such, their technique cannot be used to fine-tune an already-trained model.

Integer (INT) quantization is a promising direction due to broad hardware support [4], at the cost of narrower dynamic range. SwitchBack [38] trains vision models with up to 1B parameters from scratch, but only quantizes *two out of three matrix multiplications in the linear modules in INT8*, while retaining the third in high precision. Jetfire [41] proposes a more complex 2D block-wise quantization approach for training from scratch in INT8. Jetfire obtains good accuracy and significant end-to-end speedups ($1.4 - 1.5\times$); yet, their method was not tested for fine-tuning, and changing the entire data flow to INT8 comes with significant challenges. The recent DeepSeek FP8 training [11] adopts a similar approach to Jetfire, using blocks of size 256 and a different implementation.

We compare to SwitchBack and JetFire/DeepSeek as baselines, in the context of fine-tuning. HALO achieves similar or higher accuracy than SwitchBack, while consistently outperforming it in terms of runtime, due to executing all multiplications in low precision. Relative to JetFire/DeepSeek, we validate that FP8 training is indeed lossless, but that both methods are lossy when applied to INT8 or FP6 formats. For INT8/FP6, HALO offers higher accuracy.

Generally, efficient fine-tuning is an important workload in the context of high-quality open-weight models. Most recent work focuses on making fine-tuning more efficient by reducing memory usage [12; 27] through PEFT-style schemes [19]. HALO is completely compatible with PEFT schemes. We present results from integrations with LoRA and QLoRA, but our focus is on accelerating the actual computations during fine-tuning, which remains a less-explored area.

**Linear Layers.** Let matrices $\mathbf{W} \in \mathbb{R}^{n \times m}$, $\mathbf{X} \in \mathbb{R}^{b \times m}$ and $\mathbf{Y} \in \mathbb{R}^{b \times n}$ be the weights, inputs, and outputs of an $n \times m$ linear layer acting on an input with batch size $b$. The forward and backward passes include the following matrix multiplications, in PyTorch notation [31]:

$$\mathbf{Y} = \mathbf{X} \cdot \mathbf{W^T} \quad \text{(1a)} \qquad \mathbf{G} = \mathbf{E_Y^T} \cdot \mathbf{X} \quad \text{(1b)} \qquad \mathbf{E_X} = \mathbf{E_Y} \cdot \mathbf{W} \quad \text{(1c)}$$

where $\mathbf{G}$, $\mathbf{E_X}$, and $\mathbf{E_Y}$ are the gradients w.r.t. the weights, inputs, and outputs (where the last two are known as *errors*), respectively. The first matrix multiplication occurs during the forward pass, denoted by $\mathbf{F}$, while the last two occur during the backward pass: one for computing the weight gradients, denoted by $\mathbf{G}$, and the other for computing the errors, denoted by $\mathbf{E}$.

**Hadamard Transformations.** For a fixed dimension $\mathbf{d}$, a normalized Hadamard matrix $\mathbf{H_d}$ is an orthonormal matrix where $\mathbf{H_d H_d^T = I}$. When $\mathbf{d = 2^n}$, $\mathbf{H_d}$ is called the Walsh-Hadamard matrix; its entries are $\pm \frac{1}{\sqrt{\mathbf{d}}}$ and can be built via the recursive construction $\mathbf{H_d = H_2 \otimes H_{d/2}}$.

Assume a matrix $\mathbf{A} \in \mathbb{R}^{\mathbf{d \times d}}$. Then, a **right-hand-side Hadamard** transformation of $\mathbf{A}$ applies the linear transformation $\mathbf{h(A) = AH_d}$. When $\mathbf{d}$ is power-of-two, $\mathbf{H_d}$ is the Walsh-Hadamard matrix, and the above transformation can be done using a recursive algorithm with $O(d \log d)$ operations [45]. Following [3], when $\mathbf{d}$ is not a power of two, we factorize it as $\mathbf{d} = 2^n m$, where $\mathbf{m}$ is the size of a known Hadamard matrix. We then apply the Kronecker construction: $\mathbf{H_d = H_{2^n} \otimes H_m}$. We also use the **left-hand-side Hadamard** transformation, defined as $\mathbf{h'(A) = H_d^T A}$, which can be calculated by transposing $\mathbf{h(A^T)}$.

**Full and Parameter-Efficient Fine-Tuning.** Full fine-tuning (FFT) involves adjusting all the parameters of a pre-trained model on a downstream task, but can be infeasible for LLMs on consumer hardware. Parameter-efficient fine-tuning (PEFT) methods such as LoRA [19] focus on reducing memory usage during fine-tuning. A LoRA linear layer is parameterized by a non-trainable weight matrix $\mathbf{W} \in \mathbb{R}^{n \times m}$, and trainable components $\mathbf{U} \in \mathbb{R}^{r \times m}$ and $\mathbf{V} \in \mathbb{R}^{n \times r}$ with $r << \min(m, n)$. In the forward pass, the input tensor $\mathbf{X}$ will be passed through the frozen and low-rank weights and the output will be calculated as $\mathbf{Y = X \cdot W^T + (X \cdot U^T) \cdot V^T}$. In the backward pass, the gradient will be calculated only for the low-rank matrices. For the details of the operations, see Appendix A.3.

# 3 Method

In this section, we introduce our proposed HALO method for low-precision fine-tuning of pre-trained models. We begin by analyzing the key challenges of applying quantization during fine-tuning. Next, we present our solution to these challenges and define the levels of HALO. Finally, we discuss the details of the implementation and integration of quantized communication.

## 3.1 Design Space and Challenges of Low-Precision Fine-Tuning

Our primary goal is to apply Hadamard transformations to reduce quantization error while performing the forward and backward passes of linear layers in low precision. This approach introduces several challenges that should be addressed to achieve an optimal balance between accuracy and performance. In this (and the next) section, we outline these challenges and propose solutions for each.

**Challenge 1: Orthogonal Transformation Absorption.** QuaRot [3] eliminates the overhead of applying Hadamard transformations during inference by absorbing them into the weights of preceding layers. This is feasible when the network exclusively uses RMSNorm as its normalization module which has a distributive property: for any matrix $\mathbf{X}$ and orthogonal matrix $\mathbf{Q}$, we have $\text{RMSNorm}(\mathbf{XQ}) = \text{RMSNorm}(\mathbf{X})\mathbf{Q}$. As a result, if $\mathbf{Q}$ is absorbed into the output dimension of weight matrix in the previous layers, it will be preserved across subsequent layers—removing the need for explicit transformations during inference. This technique is known as computational invariance [2]. However, this property does not hold during the backward pass, as the derivative of RMSNorm lacks the same distributive behavior (see Appendix A.1). **Therefore, QuaRot cannot be directly extended to the backward pass to eliminate the cost of applying Hadamard transformations.**

**Challenge 2: Hadamard Combinations and Overheads.** For given matrices $\mathbf{A} \in \mathbb{R}^{m \times k}$ and $\mathbf{B} \in \mathbb{R}^{k \times n}$, our goal is to perform matrix multiplication $\mathbf{Y = A_Q B_Q}$ in low precision, where $\mathbf{A_Q}$ and $\mathbf{B_Q}$ are the quantized versions of $\mathbf{A}$ and $\mathbf{B}$, respectively. In the **No Hadamard Case** (denoted $\mathbf{Y}$), we directly apply the quantization function and compute $\mathbf{A_Q B_Q}$ without any Hadamard transformation. In the **Left Case** (denoted $\mathbf{^H Y}$), we apply a left-hand-side Hadamard transformation

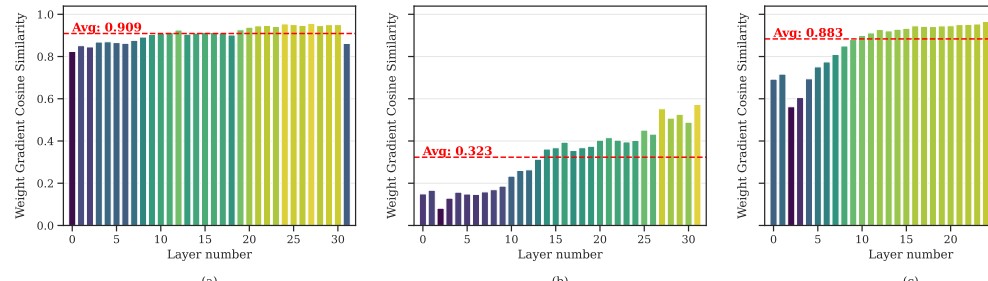

Figure 2: Cosine similarity between the weight gradients of the baseline model (BF16) and the quantized model when quantization is applied during the **(a)**: backward pass, **(b)**: forward pass, and **(c)**: forward pass with Hadamard transformation in a single fine-tuning step of LLAMA3-8Bon the GSM8K dataset. Compared to the backward pass, the forward pass is more sensitive to quantization. We improve the results by applying Hadamard transformation during the forward pass ($\overset{H}{F}$). For each case, we present the weighted average (over # parameters) for all linear modules in each layer.

of size $m$ to $\mathbf{A}$ before quantization and compute $\mathbf{H_m}(\mathbf{H_m^T A})_\mathbf{Q}\mathbf{B_Q}$[3]. In the **Right Case** (denoted $\mathbf{Y^H}$), we apply a right-hand-side Hadamard transformation of size $n$ to $\mathbf{B}$, perform $\mathbf{A_Q}(\mathbf{BH_n})_\mathbf{Q}\mathbf{H_n^T}$.

Lastly, in the **Middle Case** (denoted $\overset{H}{Y}$), we apply Hadamard transformations of size $k$ from the right to $\mathbf{A}$ and from the left to $\mathbf{B}$, computing $(\mathbf{AH_k})_\mathbf{Q}(\mathbf{H_k^T B})_\mathbf{Q}$. Any combination of these placements yields 512 possible modes, including the "No-Hadamard" case. Selecting the optimal insertion strategy is challenging because: (1) we aim to minimize the number of Hadamard transformations to reduce computational overhead; and (2) we must carefully place them to suppress outliers and maintain stability during low-precision training (see Appendix A.7 for an ablation study over different combinations). **Naively evaluating all combinations of Hadamard placements for each fine-tuning task is clearly impractical.**

**Challenge 3: Communication and Memory Issues.** Fine-tuning large models typically requires multiple GPUs, which must communicate over a network to perform operations like weight gathering (AllGather) and gradient aggregation (AllReduce). These communication steps introduce significant overhead during training. Additionally, storing activations for the backward pass (especially with large batch sizes) can become a major memory bottleneck. **Therefore, HALO has to also consider communication and memory efficiency constraints in the method design.**

## 3.2 HALO Solutions for Fine-tuning

As discussed in the previous section, quantized fine-tuning presents several challenges. In this section, we present the main steps of designing HALO to address the above challenges during fine-tuning.

**Step 1: Sensitivity Analysis of Forward vs. Backward Quantization.**

We begin by identifying the most quantization-sensitive matrix multiplications during the forward and backward passes. While it is known that activations ($\mathbf{X}$) and error gradients ($\mathbf{E_Y}$) exhibit outliers in their columns and rows, respectively [3; 39; 7], it remains unclear which specific matrix multiplications are most vulnerable to quantization effects. To address this, we study the cosine similarity between the weight gradients of the baseline model (no quantization) and the quantized model when quantization is applied during the forward pass or the backward pass.

Figure 2 shows that the computed weight gradients have significantly lower cosine similarity compared to the baseline model when quantization is applied during the forward pass. Specifically, when we only quantize Equations (1b–1c) in the backward pass, the average cosine similarity is 0.909. In contrast, when quantization is applied to the forward pass in Equation (1a), while leaving the backward pass unchanged, the average cosine similarity drops to 0.323. This drop may result from larger quantization errors in the activations, potentially caused by outliers, or from errors introduced during the network loss computation after the forward pass is quantized.

---

[3]We apply another left-hand-side Hadamard transformation on the output of the low-precision matrix multiplication to (approximately) cancel the effect of this transformation.

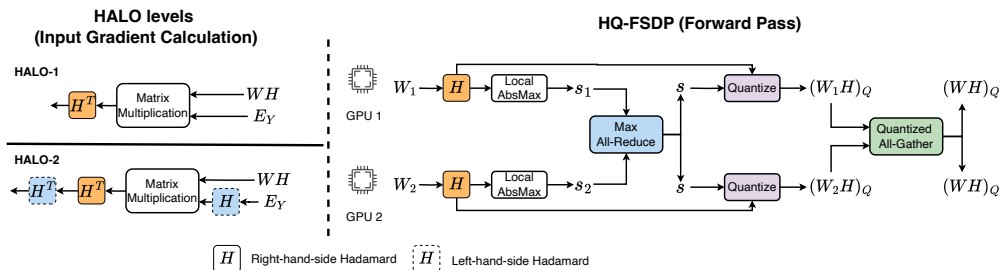

Figure 3: **Left**: Hadamard transformations in HALO levels during the input gradient calculation. Hadamard transformations are already applied to the weights during the forward pass. **Right**: Forward pass in HQ-FSDP for two GPUs: Each GPU performs a right-hand side Hadamard transformation and computes the absolute maximum (AbsMax) $s_i$ over its local weight shard $W_i$. Then, all GPUs participate in an `All-Reduce` operation and compute the global maximum absolute value $s$. Each GPU uses this global value to quantize its own weight shard. Finally, an `All-Gather` operation is performed on the quantized shards $(W_i H)_Q$. All GPUs use the same scales to quantize the weights in the backward pass after applying Hadamard transformation.

We conclude that *mitigating outliers is key during the forward pass*, and we can recover most of the averaged cosine similarity (0.883) by applying Hadamard transformations during the forward pass on the weights and inputs.

**Step 2: Memory- and Communication-Efficient Backward Pass.**

In Step 1, we apply a right-hand-side Hadamard transformation to both weights and activations. Due to the orthogonality of the Hadamard matrix, these transformations cancel out in Equation (1a). In the next step, we retain the quantized transformed weights $((\mathbf{WH})_{\mathbf{Q}})$ and activations $((\mathbf{XH})_{\mathbf{Q}})$ for the backward pass and perform the computations defined in Equations (1b–1c) using these tensors. This ensures consistency between the forward and backward passes, as both operate on the same (quantized) inputs and weights, while also enabling efficient reduction in storage and communication.

To recover the true weight and input gradients in Equations (1b–1c), an additional right-hand-side Hadamard transformation must be applied to the outputs of the corresponding matrix multiplications in the backward pass, introducing two extra Hadamard transformations into the overall scheme.

**Step 3: Outlier Mitigation for Output Gradients (Errors) in the Backward.**

With the first two steps, we can stabilize fine-tuning at precisions with moderate dynamic ranges (such as FP6). However, for narrower dynamic ranges (e.g., INT8), it becomes necessary to mitigate outliers in the error gradients during the backward pass.

Figure 1-Left shows that, unlike activation tensors $\mathbf{X}$, in the error tensors $\mathbf{E_X}$ the outliers are propagated *across columns* (channel/feature dimension). Then, using a right-hand-side Hadamard transformation does not solve the outlier issue as such transformation rotates the input rows (Figure 1-Middle). Thus, we apply *left-hand-side Hadamard transformation*, or $\mathbf{h}'(\mathbf{E_Y})$, which mitigates the error outliers by rotating the column vectors (Figure 1-Right).

We avoid applying the Hadamard transformation during weight gradient computation because (1) computing input gradients is more sensitive to quantization, as these gradients are propagated to previous layers, and (2) performing a left-hand-side Hadamard transformation is more computationally expensive due to the need for two transpositions.

### 3.3 The HALO Method

**Full Fine Tuning.** Based on the above steps, we define two levels within HALO, corresponding to data formats with moderate and narrow dynamic ranges, respectively as follows:

1. At the first level, denoted by HALO-1($\overset{\mathbf{H}}{\mathbf{F}}, \mathbf{E^H}, \mathbf{G^H}$), we employ the middle case during the forward pass to mitigate issues with weight gradients (Figure 2c). Specifically, right-hand-side Hadamard transforms are applied to both input and weight matrices prior to quantization. This introduces two implementation concerns: 1) *Low-Precision Communication*: We apply

a right-hand-side Hadamard transform to the weights in Equation (1c) to facilitate integration with HQ-FSDP and Activation Checkpointing. 2) *Memory Reduction*: To improve memory efficiency, we apply a right-hand-side Hadamard transform to the input during the backward pass in Equation (1b). This allows us to reuse the quantized inputs from the forward pass, eliminating the need to keep the high-precision inputs in memory.

2. Finally, to mitigate outliers in errors, we define the highest level by applying a left-hand-side Hadamard transformation to $\mathbf{E}$, denoted by HALO-2($\overset{\text{H}}{\mathbf{F}}, {}^{\text{H}}\mathbf{E}^{\text{H}}, \mathbf{G}^{\text{H}}$).

The main difference in the HALO levels is applying Hadamard transformations during the error calculations. Table 3 (in Appendix A.2) summarizes the key modifications introduced to the matrix multiplication computations across different HALO levels. The primary distinction between HALO-1 and HALO-2 is the application of a left-hand-side Hadamard transformation to the errors during input gradient calculation, as shown in Figure 3-Left.

**Parameter Efficient Fine Tuning.** To accelerate parameter-efficient fine-tuning (PEFT), we apply ($\overset{\text{H}}{\mathbf{F}}, {}^{\text{H}}\mathbf{E}^{\text{H}}, \mathbf{G}$) on the large matrices while maintaining low-rank operations in high precision due to their inherent efficiency. For the weight quantization, we apply a right-hand-side Hadamard transformation and quantize the frozen weights once and store the quantized weights before fine-tuning. We then apply a single right-hand-side Hadamard transformation during the forward pass on the inputs and two Hadamard transformations during the backward pass on the errors. Finally, we keep the gradient calculation in high precision as it uses only low-rank matrix multiplication. We denote this variant scheme by **HALO$_{\text{PEFT}}$**. We present the details of the Equations in Appendix A.3.

### 3.4 HQ-FSDP: HALO Quantized Communication and Memory Reduction

In this section, we describe how HALO leverages quantized Fully Sharded Data Parallel (FSDP) to reduce communication overhead, and quantized activation storage to minimize memory usage.

**FSDP Integration.** Fully Sharded Data Parallel (FSDP) [48] is a common distributed training strategy for LLM fine-tuning. In FSDP, model weights are *sharded* (i.e., distributed) across multiple GPUs. Whenever necessary, a subset of the weights (typically corresponding to a transformer block) is *all-gathered*, enabling every GPU to have the full weight subset required for an operation. This communication occurs before the forward and backward passes of each block. After the backward pass, gradients are *reduce-scattered*, ensuring that each process maintains the averaged gradient for its own shard (Figure 3-Right).

Since HALO requires only the quantized weights for performing the $\mathbf{F}$ and $\mathbf{E}$ operations, it allows for low precision all-gather communications. We refer to this approach as Hadamard-Quantized FSDP (HQ-FSDP) and implement it as follows: (a) if necessary, each process applies a right-hand Hadamard transformation to its shard (depending on the HALO level implemented), (b) each process computes a local quantization scale for its shard, (c) the scales are max-reduced to ensure that every process has the global scale per weight matrix, (d) each process quantizes its shard with the global scales, and (e) the low-precision quantized weights are communicated. This approach significantly reduces communication overhead while distributing the quantization and, possibly, the Hadamard transformation overhead across processes. Notably, the global scales calculated during the forward pass are reused in the backward pass, skipping steps (b) and (c) above. See Appendix A.5 for details.

FSDP is often combined with Activation Checkpointing (AC), which saves only selected activations (*checkpoints*, e.g., before and after transformer blocks) during the forward pass, reducing memory usage. Before the backward pass of each block, intermediate activations are recomputed via a second forward pass. With AC, FSDP communicates each weight only once during the backward pass, reusing it for both the second forward pass and the backward pass. Thus, when using HQ-FSDP with AC, applying the same Hadamard transformation to the weights in both $\mathbf{F}$ and $\mathbf{E}$ operations is essential to ensure communication speedup.

**Activation Quantization For Memory Reduction.** Xi et al. [40] have shown that up to 40% of memory is used to store activations during training in LLAMA-style models. This is because the activation tensor $\mathbf{X}$ in Equation (1a) must be stored and reused in Equation (1b) during the backward pass. We always store the quantized version of $\mathbf{X}$ for the backward pass to address this issue. When we apply a right-hand Hadamard operation on $\mathbf{X}$ during the forward pass, we need to apply another right-hand Hadamard operation on the output of Equation (1b) to ensure identical computation.

## 4 Experimental Validation

We implement HALO in `PyTorch` [31] based on the the `llm-foundry` codebase [25] for FFT, and the standard HuggingFace PEFT library for **HALO$_{\text{PEFT}}$**. We use **tensor-wise symmetric quantization** (a single scale for the entire tensor) for all data types, and Round-to-Nearest (RTN) quantization across all our experiments. We implement our own low-precision matrix multiplications using the `CUTLASS` library [29] for all linear modules (except for the LM head and embeddings) and keep the rest of the model in the original precision (BF16) during fine-tuning. For outlier mitigation, we adapt efficient Hadamard CUDA kernels [10]. We use E4M3 for the 8-bit floating-point format; after comparing accuracies, we chose the E3M2 format for FP6 as the more accurate variant.

**Model, Tasks, and Hyper-parameters.** For FFT, we evaluate HALO on LLAMA3-8B [13] as well as large scale Qwen (14B and 32B) models [44] (in Appendix A.13), following published fine-tuning recipes [26; 27] for both FFT and PEFT. For both FFT and PEFT, we consider three standard datasets: 1) ViGGO [20], with 5.1k training and 1.08k test samples, 2) Grade-School Math (GSM8k) [8], with 7.74k training and 1.32k test samples, and 3) SQL generation [49; 46], with 30k training and 1k test samples. These datasets are particularly interesting because they are either highly-specialized (like SQL and ViGGO), or the pre-trained model has low few-shot accuracy (like GSM8K), making fine-tuning necessary. We use the same hyperparameters as BF16 fine-tuning, detailed in Appendix A.6. Each experiment is repeated with 5 seeds; we report the mean and standard error.

**Baselines.** For full fine-tuning, we compare HALO against three main baselines: BF16; `SwitchBack` [38], and `Jetfire` [41], as described in Section 2. `SwitchBack` uses row-wise quantization for the inputs and errors during the forward and backward passes and performs weight gradient calculation in 16 bits. `Jetfire` applies 2D block-wise quantization with a block size of $32 \times 32$ on the linear layers and use symmetric AbsMax for quantization. Notably, for fair comparison, we maintain this scheme's data flow in 16 bits and do not quantize the activation functions in our `Jetfire` baseline, as this aspect is orthogonal to our scheme. Finally, we compare HALO with the standard PEFT methods LoRA [19] with rank $r = 8$.

### 4.1 Low-Precision Full Fine-Tuning

**Integer (INT8) Quantization.** The first row in Table 1 summarizes HALO results across three tasks, where weights, activations, and errors are quantized using INT8 quantization. We utilize the second level of our scheme, denoted by HALO-2($\overset{\text{H}}{\text{F}}$, $^{\text{H}}\text{E}^{\text{H}}$, $\text{G}^{\text{H}}$), for integer quantization and compare our results against BF16 (no quantization), `SwitchBack`, and `Jetfire`. Across all tasks, HALO and `Jetfire` achieve relative accuracy within 1% of the BF16 model. However, HALO employs tensor-wise quantization for all tensors and has lower variance across different seeds. Additionally, our version of `Jetfire` retains the data flow in BF16, making it more accurate than the original implementation. Finally, `SwitchBack` fails to recover accuracy on both the ViGGO and GSM8K datasets, despite using row-wise and column-wise quantization for inputs and weights, along

| Format | Method | GSM8k | ViGGO | SQL |
|--------|--------|-------|-------|-----|
| BF16 | Baseline | $69.3 \pm 0.5$ | $94.0 \pm 0.3$ | $79.9 \pm 0.5$ |
| INT8 | SwitchBack | $64.0 \pm 0.7$ | $61.0 \pm 28.9$ | $79.6 \pm 0.8$ |
| | Jetfire | $68.2 \pm 0.6$ | $93.6 \pm 0.4$ | $80.2 \pm 0.6$ |
| | HALO-2 | $68.3 \pm 0.1$ | $93.8 \pm 0.1$ | $80.1 \pm 0.3$ |
| FP6 | SwitchBack | $62.7 \pm 0.7$ | $93.4 \pm 0.5$ | $80.1 \pm 0.5$ |
| | Jetfire | $62.8 \pm 1.1$ | $92.9 \pm 0.2$ | $79.7 \pm 0.5$ |
| | HALO-1 | $66.5 \pm 0.2$ | $93.6 \pm 0.4$ | $80.2 \pm 0.3$ |
| FP8 | SwitchBack | $69.1 \pm 0.3$ | $93.2 \pm 0.2$ | $80.4 \pm 0.3$ |
| | Jetfire | $69.3 \pm 0.4$ | $93.1 \pm 0.3$ | $80.2 \pm 0.3$ |
| | No-HALO | $69.2 \pm 0.2$ | $93.2 \pm 0.2$ | $80.3 \pm 0.2$ |

Table 1: Single-epoch accuracy results of fine-tuning with 8-bit (INT8/FP8) and 6-bit (FP6) quantizations. We use HALO-2($\overset{\text{H}}{\text{F}}$, $^{\text{H}}\text{E}^{\text{H}}$, $\text{G}^{\text{H}}$) for integer quantization and HALO-1($\overset{\text{H}}{\text{F}}$, $\text{E}^{\text{H}}$, $\text{G}^{\text{H}}$) for FP6 (E3M2), while we use no Hadamard for FP8 (E4M3), denoted by No-HALO.

with high-precision weight gradient calculations. The SQL dataset appears to be "easier", as all schemes match the baseline after 1 epoch of FFT.

**Floating-Point Quantization.** Next, we study the use of floating-point representation during fine-tuning by applying FP6 (E3M2) and FP8 (E4M3) quantization. For FP6, we utilize the first level, denoted by HALO-1($\overset{\text{H}}{\text{F}}$, $\text{E}^{\text{H}}$, $\text{G}^{\text{H}}$), where we apply the Hadamard transform on the forward pass and use the Hadamard-transformed weights and inputs during the backward calculation. On GSM8K, both `SwitchBack` and `Jetfire` have ~6.5% accuracy drop, while HALO-1 shows a 2.8% accuracy loss compared to the BF16 model. On ViGGO, HALO experiences a 0.4% accuracy drop, whereas

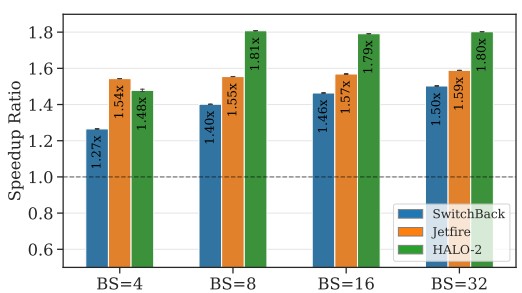 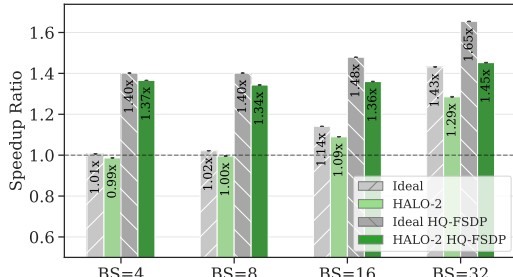

(a) Linear layer (of size $4096 \times 4096$).      (b) Three consecutive transformer blocks (with/without HQ-FSDP).

Figure 4: INT8 forward + backward speedups (over BF16) for $512$ sequence length in LLAMA3-8B across batch sizes (BS). HALO-2 refers to HALO-2($\overset{\text{H}}{\text{F}}$, $^{\text{H}}\text{E}^{\text{H}}$, $\text{G}^{\text{H}}$).

SwitchBack and Jetfire show drops of 0.6% and 1.1%, respectively. Similar to integer quantization, all methods achieve accuracy within 1% of the baseline on the SQL dataset. Although the peak performance of FP6 is the same as FP8 [30], FP6 provides higher memory and communication reduction with quantized activations and HQ-FSDP (see Section 3). We present the accuracy results for HALO$_{\text{PEFT}}$ in Appendix A.3.

## 4.2 Speedup Analysis

We now evaluate the runtime improvements achieved with HALO. Speedups are measured on RTX 4090 GPUs with locked clocks, to reduce variance, for: a single linear layer and for end-to-end training. We compare different precisions and HALO levels against the BF16 base case and other baselines. Unless mentioned, we fix the sequence length at 512 and control the input size using the batch size. The layer-wise and block-wise speedups are averaged over $100$ (+20 warm-up) and $30$ (+10 warm-up) runs, respectively.

**Linear Layer.** First, we consider a single linear layer of size $4096 \times 4096$, matching the majority of linear modules in the LLAMA3-8B model. We evaluate our most accurate INT8 quantization scheme, HALO-2($\overset{\text{H}}{\text{F}}$, $^{\text{H}}\text{E}^{\text{H}}$, $\text{G}^{\text{H}}$), relative to the SOTA Jetfire and SwitchBack methods. Since Jetfire uses an INT8 data flow, we consider a setup where its inputs, outputs, and errors are in INT8, bypassing its quantization overhead: this is an "idealized" version of Jetfire to ensure a fair comparison. Figure 4-(a) illustrates the speedups relative to the baseline BF16 implementation. HALO-2 attains additional 20-40% speedups compared to SwitchBack, primarily due to the higher precision matrix multiplication for weight gradient calculations in SwitchBack. Additionally, for batch sizes greater than 4, HALO-2 has at least 20% more speedup than Jetfire, as Jetfire's complex 2D block-wise quantization introduces significant overhead during both quantization and dequantization. However, for a batch size of 4, Jetfire is slightly faster than HALO-2 due to the overhead from Hadamard transformations and transpositions. Switching HALO to a low-precision data flow would increase speedups; we leave this for future work.

**The Effect of HQ-FSDP.** To evaluate the impact of HQ-FSDP across different batch-sizes, we run forward and backward passes over three consecutive transformer blocks that fit in GPU memory at BS=32—using four GPUs with FSDP enabled. Figure 4-(b) reports speedups with and without quantized communication. HQ-FSDP achieves $1.37\times$ to $1.43\times$ speedups, delivering 10-40% higher speedups compared to 16-bit communication. The improvement is more significant at smaller batch sizes (e.g. BS=4), as communication becomes the primary bottleneck for training.

**End-to-End Speedups.** Finally, in Table 2, we compare the end-to-end fine-tuning speedups for LLAMA3-8B when we use INT8 HALO-2 and FP8 without Hadamard transformation, denoted by No-HALO, both of which are near-lossless for the corresponding precisions. Using four GPUs, we can fit only four samples into GPU memory, whereas with eight GPUs, we can fit up to eight samples, each with

| NVIDIA | 4x GPUs | | 8x GPUs | |
|---|---|---|---|---|
| (RTX-4090) | BS=4 | BS=8 | BS=4 | BS=8 |
| INT8 (HALO-2) | $1.35\times$ | OOM | $1.41\times$ | $1.41\times$ |
| FP8 (No-HALO) | $1.36\times$ | OOM | $1.41\times$ | $1.39\times$ |

Table 2: End-to-end speedups one epoch LLAMA3-8B full fine-tuning with best performing HALO level using INT8 and FP8.

512 tokens. In the former case, No-HALO and HALO-2 achieve speedups of $1.35\times$ and $1.36\times$, respectively. However, the speedups are higher with eight GPUs, reaching up to $1.41\times$, mainly due to increased communication overhead in this configuration. All experiments were conducted in a realistic setting for both four and eight GPUs, with HQ-FSDP and Activation Checkpointing enabled. W In addition, in Appendix A.9, we provide inference speed results for the model which is fine-tuned using HALO, showing that the Hadamard transformations have minimal impact on inference performance.

## 5    Conclusion

We introduced HALO, an LLM fine-tuning scheme that performs all matrix multiplications in lower-precision, leveraging Hadamard transforms to mitigate outliers. HALO uses simple tensor-wise quantization for all weights, inputs, and errors, utilizes low-precision communication (HQ-FSDP), and reduces memory usage by storing quantized activations. For INT8, HALO achieves up to a $1.36\times$ end-to-end speedup which is comparable to the $1.41\times$ speedup of FP8 without any Hadamard transformations. This holds during fine-tuning of LLAMA3-8B on four and eight commodity GPUs, while maintaining accuracy close to the baseline. In future work, we plan to investigate HALO for pre-training accurate models from scratch, and extend it for additional GPU hardware types.

## Acknowledgements

This project has received funding from the European Research Council (ERC) under the European Union's Horizon 2020 program (grant agreement PSAP, No. 101002047. This research also obtained funding from the "UrbanTwin: An urban digital twin for climate action: Assessing policies and solutions for energy, water and infrastructure" project, funded by the ETH-Domain Joint Initiative program in the Strategic Area Energy, Climate and Sustainable Environment.

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

# A Technical Appendices and Supplementary Material

## A.1 RMSNorm Backward Pass

For every vector $\mathbf{x} \in \mathbf{R}^\mathbf{d}$, we define

$$f(x) = \frac{x}{\|x\|} = \frac{x}{\sqrt{\sum_{i=1}^{d} x_i^2}} = \frac{x}{\sqrt{x^\top x}}. \tag{2}$$

Given a matrix $\mathbf{X}$, the $\mathbf{RMSNorm}(\mathbf{X})$ applies Equation (2) on each row of $\mathbf{X}$ in the forward pass. For a given orthogonal transformation matrix $\mathbf{Q}$, we will have $\mathbf{f}(\mathbf{Qx}) = \mathbf{Qf}(\mathbf{x})$ as $\|\mathbf{Qx}\| = \sqrt{\mathbf{x^T Q^T Qx}} = \|x\|$, which is known as the *distributive property*. Using this fact, Ashkboos et al. [2] showed that any orthogonal transformation $\mathbf{Q}$ can be fused to the weights of the previous layer, and RMSNorm will preserve it because of the distributive property above. This approach eliminates the overhead of applying Hadamard transformations during the forward pass.

Now, consider the derivative of $\mathbf{f}(\mathbf{x})$ which is used during the backward pass of fine-tuning. For a given input vector $\mathbf{x} \in \mathbf{R}^\mathbf{d}$, we will have

$$\frac{\partial}{\partial x} \left( \frac{x}{\|x\|} \right) = \frac{1}{\|x\|} \left( I - \frac{xx^\top}{\|x\|^2} \right), \tag{3}$$

which **does not** have the above distributive property. This shows that one cannot use the fusion idea [3; 2; 23] to merge the Hadamard transformations into the previous weight matrix during the backward pass, especially as we need to apply left-hand-side Hadamard transformations on the errors (see Figure 1).

## A.2 HALO Levels for FFT

We summarize all the HALO levels for fine-tuning of a linear layer in Table 3. We use tensor-wise symmetric quantization function $\mathbf{Q}$ for all data types (input, weights, and errors).

| Method | Forward Calculation ($\mathbf{F}$) | Input Gradient Calculation ($\mathbf{E}$) | Weight Gradient Calculation ($\mathbf{G}$) | Notes |
|---|---|---|---|---|
| No-HALO | $\mathbf{X_Q W_Q^T}$ | $(\mathbf{E_Y})_\mathbf{Q} \mathbf{W_Q}$ | $(\mathbf{E_Y^T})_\mathbf{Q} \mathbf{X_Q}$ | 1. Suitable for wide ranges (e.g., FP8). |
| HALO-1($\mathbf{\overset{H}{F}}, \mathbf{E^H}, \mathbf{G^H}$) | $(\mathbf{XH})_\mathbf{Q}(\mathbf{WH})_\mathbf{Q}^\mathbf{T}$ | $(\mathbf{E_Y})_\mathbf{Q}(\mathbf{WH})_\mathbf{Q} \mathbf{H^T}$ | $(\mathbf{E_Y^T})_\mathbf{Q}(\mathbf{XH})_\mathbf{Q} \mathbf{H^T}$ | 1. Outlier mitigation during the forward pass. 2. Easy integration with FSDP with AC. 3. Quantized activation for memory reduction. 4. Suitable for moderate ranges (e.g., FP6). |
| HALO-2($\mathbf{\overset{H}{F}}, \mathbf{{}^H E^H}, \mathbf{G^H}$) | $(\mathbf{XH})_\mathbf{Q}(\mathbf{WH})_\mathbf{Q}^\mathbf{T}$ | $\mathbf{H^T}(\mathbf{HE_Y})_\mathbf{Q}(\mathbf{WH})_\mathbf{Q}\mathbf{H^T}$ | $(\mathbf{E_Y^T})_\mathbf{Q}(\mathbf{XH})_\mathbf{Q}\mathbf{H^T}$ | 1. Most accurate scheme. 2. Suitable for narrow ranges (e.g., INT8). |

Table 3: HALO levels for full fine-tuning (FFT). We use the quantization function $\mathbf{Q}$ for quantizing different data types and perform the computation with low precision. AC stands for Activation Checkpointing.

## A.3 LoRA-style Linear Module and HALO for PEFT

A LoRA linear module includes the following operations:

$$\mathbf{Y} = \mathbf{X} \cdot \mathbf{W^T} + (\mathbf{X} \cdot \mathbf{U^T}) \cdot \mathbf{V^T}, \tag{4}$$

$$\mathbf{E_X} = \mathbf{E_X^{UV}} + \mathbf{E_X^W} = (\mathbf{E_Y} \cdot \mathbf{V}) \cdot \mathbf{U} + \mathbf{E_Y} \cdot \mathbf{W}, \tag{5}$$

$$\mathbf{G_V} = \mathbf{E_Y^T} \cdot (\mathbf{X} \cdot \mathbf{U^T}), \tag{6}$$

$$\mathbf{G_U} = (\mathbf{V^T} \cdot \mathbf{E_Y^T}) \cdot \mathbf{X}, \tag{7}$$

where $\mathbf{G_U}$ and $\mathbf{G_V}$ are the gradients of $\mathbf{U}$ and $\mathbf{V}$, respectively. Since low-rank operations are fast, our goal is to quantize operations not involving the the low-rank matrices $\mathbf{U}$ and $\mathbf{V}$.

As mentioned in Section 3, we integrate HALO with PEFT by quantizing the majority of the computation while retaining the low-rank computations in high precision. In this case, the Equations (4-5) will become

$$\mathbf{Y} \approx (\mathbf{XH})_{\mathbf{Q}} \cdot (\mathbf{WH})_{\mathbf{Q}}^{\mathbf{T}} + (\mathbf{XU}^{\mathbf{T}}) \cdot \mathbf{V}^{\mathbf{T}}, \tag{8}$$

$$\mathbf{E_X} \approx \mathbf{E_X^{UV}} + \mathbf{H} \cdot (\mathbf{H^T E_Y})_{\mathbf{Q}} \cdot (\mathbf{WH})_{\mathbf{Q}} \mathbf{H^T}. \tag{9}$$

We show the above scheme by **HALO$_{\text{PEFT}}$**.

Next, we compare HALO$_{\text{PEFT}}$ against LoRA [19], on GSM8K, SQL, and ViGGO for FP8, FP6 and INT8 in Table 4. For INT8 and FP8, HALO$_{\text{PEFT}}$ is always within the standard deviation of baseline. On GSM8K, FP6 has approximately a 2% accuracy degradation, showing the challenge of recovering high-precision accuracy with only 6 bits. For ViGGO and SQL, both INT8 and FP6 recover accuracy with at most a 0.5% average accuracy drop.

Table 4: Single-epoch accuracy comparison between LoRA [19] and HALO$_{\text{PEFT}}$(ours). We use rank $r = 16$ for adapters and apply FP6 (E3M2) and INT8 for quantization.

| Precision | Method | GSM8k | ViGGO | SQL |
|-----------|--------|-------|-------|-----|
| BF16 | LoRA | $69.4 \pm 0.8$ | $94.1 \pm 0.2$ | $80.0 \pm 0.4$ |
| INT8 | | $69.0 \pm 0.5$ | $93.4 \pm 0.7$ | $79.9 \pm 0.4$ |
| FP6 | HALO$_{\text{PEFT}}$ | $67.3 \pm 0.6$ | $93.6 \pm 0.7$ | $79.9 \pm 0.5$ |
| FP8 | | $69.4 \pm 1.0$ | $94.2 \pm 0.6$ | $80.0 \pm 0.3$ |

## A.4 Hadamard Effect

Figure 5 illustrates the effect of applying Hadamard transformations to the right and left sides of the activations (input matrices), errors (gradients with respect to the output), and weights. For the activations, applying Hadamard on the right side effectively removes outliers, whereas for the errors, a left-side Hadamard transformation is necessary to eliminate outliers, as also noted in Figure 1.

## A.5 HQ-FSDP Details

Here we discuss some details about our HQ-FSDP implementation. **Master Weights.** In HALO, weights are maintained in BF16. Accordingly, HQ-FSDP stores the parameters in BF16 and applies quantization before communication. **Forward vs. Backward.** Since the weights remain unchanged between the forward and backward passes, the quantized shard could be stored during the forward pass in each process and only communicated during the backward pass. This approach eliminates the need for a second quantization and potential Hadamard transform during the backward pass. However, it introduces additional memory overhead, as the quantized weights must be stored alongside the master BF16 weights. Additionally, since the quantization and Hadamard transform on the weights are distributed across FSDP processes, their overhead is relatively small. Instead, we adopt an intermediate approach: save only the global quantization scales during the forward pass and recompute the quantization and Hadamard transform during the backward pass. **FSDP Wrapping Policy.** Following standard practice, we wrap each transformer block with an FSDP module (FSDP communications happen before each transformer block). However, for HQ-FSDP, we skip the layer-norm modules (keeping full weights in every process) as we do not intend to quantize them. Our experiments show that this only marginally increases memory usage and does not change runtime. **Distributed Hadamard Transformation.** When the scheme requires a right Hadamard transformation, HQ-FSDP applies it in a distributed way; process $i$ performs $\mathbf{W}_i \mathbf{H}$ where $\mathbf{W_i}$ denotes the $i$'th shard. However, this requires the weight matrix to be sharded by row, i.e., each row should entirely reside within a single shard. As FSDP requires equally sized shards to be fast, we dynamically insert small dummy parameter tensors of carefully chosen sizes when needed. This guarantees row-aligned sharding without compromising performance.

## A.6 Hyper-Parameters

In all experiments, we tune the hyper-parameters on the base BF16 tasks, and re-use the same values for low-precision training. We always perform single-epoch experiments using the AdamW optimizer

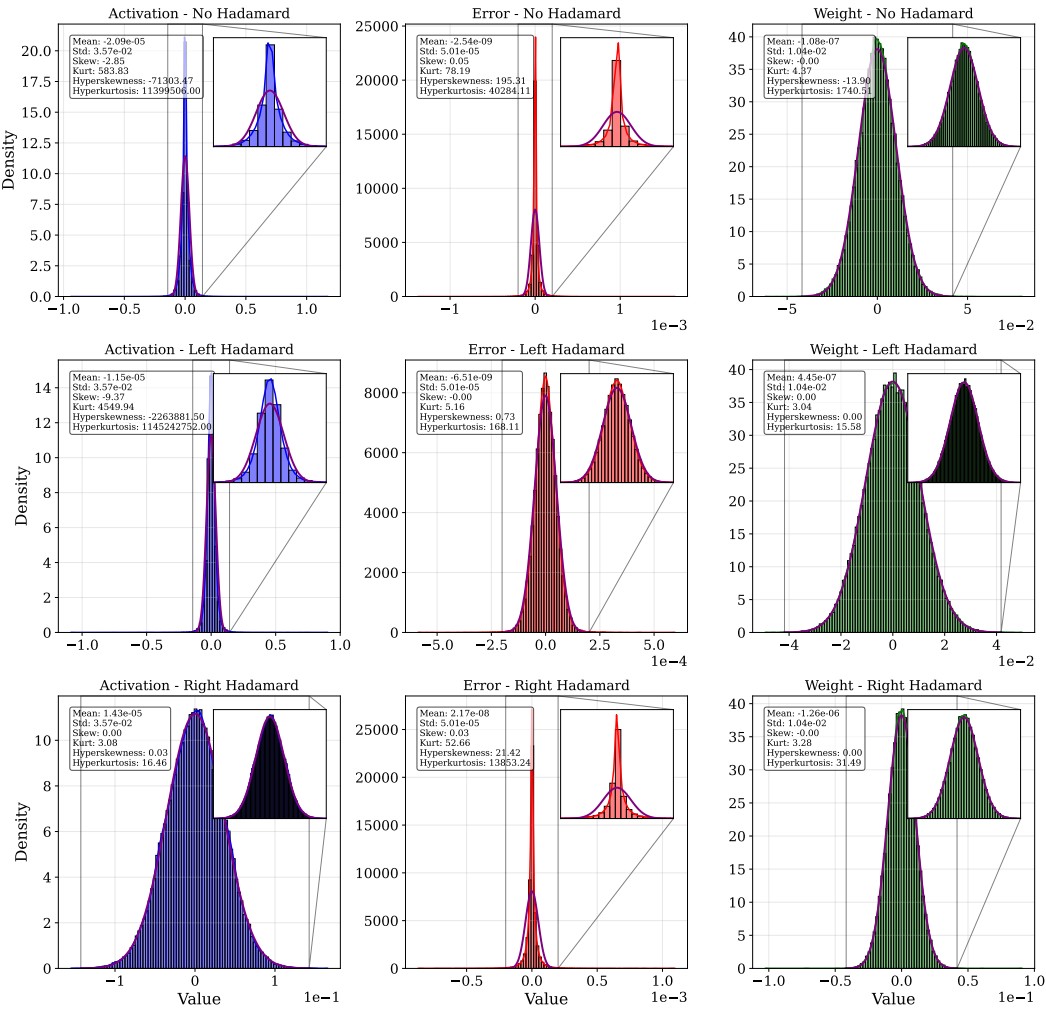

Figure 5: The effect of applying Hadamard transformations on the left and right hand sides of the the activations, errors, and weights.

with $\beta_1 = 0.9$, $\beta_2 = 0.999$, and a linear learning rate warm-up of 20 steps. The batch size and sequence length are fixed at 32 and 512. For FFT, we choose learning rates $4 \times 10^{-5}$, $6 \times 10^{-6}$, and $3 \times 10^{-5}$ for ViGGO, GSM8k, and SQL, respectively, and for PEFT LoRA experiments, we choose the learning rate $6 \times 10^{-4}$ and LoRA rank of 16 for all datasets. These learning rates were found to be the best using a grid search within the range $\left[10^{-6}, 10^{-3}\right]$ of 20 uniform log-linearly separated grid points, trained and evaluated using the non-quantized BF16 training precision.

## A.7 Ablation study on Hadamard schemes

Figure 6 presents various combinations of applying Hadamard transformations during the forward and backward passes. As shown in Figure 6-Left, applying Hadamard in the forward pass is crucial. For the backward pass, Figure 6-Right indicates that HALO-1($\overset{\text{H}}{\text{F}}$, $\text{E}^{\text{H}}$, $\text{G}^{\text{H}}$) and HALO-2($\overset{\text{H}}{\text{F}}$, $^{\text{H}}\text{E}^{\text{H}}$, $\text{G}^{\text{H}}$) are the optimal configurations, balancing the minimization of Hadamard transformations with system-level considerations such as memory usage and communication compression.

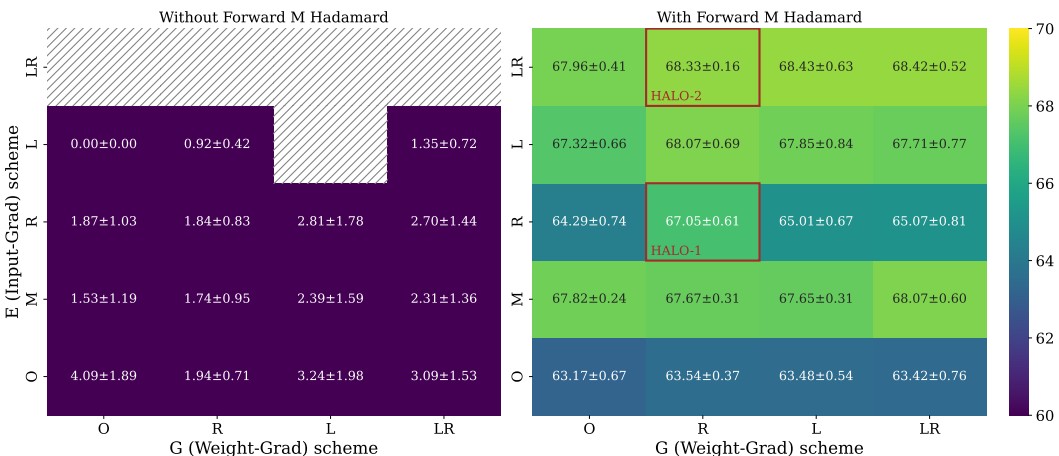

Figure 6: Different combinations of applying Hadamard transformations during the forward and backward passes of training LLAMA3-8B on GSM8K using INT8 precision. Here, "L" denotes applying a Hadamard transformation on the left-hand side, "R" indicates applying it on the right-hand side, and "M" refers to the middle case. "LR" represents applying Hadamard transformations on both the left and right sides, while "O" denotes the case where no Hadamard transformation is applied.

## A.8    Ablation Study on HALO Levels

One interesting question concerns the comparison between the different levels of HALO introduced in Section 3.3, that is HALO-0($\mathbf{F}, \mathbf{E}, \mathbf{G}$), HALO-1($\overset{H}{\mathbf{F}}, \mathbf{E^H}, \mathbf{G^H}$), and HALO-2($\overset{H}{\mathbf{F}}, {}^{H}\mathbf{E^H}, \mathbf{G^H}$) used during the fine-tuning of FP8, FP6, and INT8, respectively.

Table 5 shows the effect of using different levels for INT8 and FP6, illustrating the natural finding that a higher HALO level results in higher final test accuracy. For INT8 precision, HALO-0 fails to recover accuracy, leading to an approximate 40% drop in accuracy (on average) on our datasets. At the next level, HALO-1 recovers approximately 37% of the above accuracy gap by applying right-hand-side Hadamard transformations on the weights and inputs during both the forward and backward passes. Finally, HALO-2 applies left-hand-side Hadamard transformations on the errors, achieving within 1% of the BF16 accuracy. For FP6 precision, although HALO-2 achieves higher accuracy on the GSM8K dataset, believe HALO-1 provides a better trade-off between accuracy and the number of Hadamard transformations.

Table 5: The accuracy effects of using different HALO levels within each quantization precision. The selected level for each precision is presented with **bold** text. We exclude FP8 experiments as HALO-0 recovers the BF16 accuracy.

| Precision | Method | GSM8k | ViGGO | SQL |
|---|---|---|---|---|
| BF16 | Baseline | $69.26 \pm 0.51$ | $94.02 \pm 0.29$ | $79.83 \pm 0.49$ |
| FP6 | HALO-0 | $62.32 \pm 0.65$ | $92.92 \pm 0.73$ | $79.24 \pm 0.36$ |
|  | HALO-1 | $66.54 \pm 0.22$ | $93.56 \pm 0.38$ | $80.20 \pm 0.26$ |
|  | HALO-2 | $67.42 \pm 0.99$ | $93.56 \pm 0.38$ | $80.00 \pm 0.62$ |
| INT8 | HALO-0 | $4.50 \pm 1.03$ | $55.98 \pm 20.89$ | $74.73 \pm 0.45$ |
|  | HALO-1 | $62.27 \pm 0.64$ | $93.23 \pm 0.45$ | $79.43 \pm 0.59$ |
|  | HALO-2 | $68.15 \pm 0.08$ | $93.79 \pm 0.08$ | $80.12 \pm 0.31$ |

## A.9    HALO Inference Speedups

We present HALO as a low-precision fine-tuning method. However, it can be presented as a QAT scheme where the inference of the fine-tuned model will be done in low precision. To this end, following QuaRot [3], we fuse the Hadamard transformations into the previous linear modules and just apply two Hadamards before `out-projection` and `down-projection` layers in the Attention and MLP modules. Table 6 shows the inference speedups of a single Transformer block when we

use HALO for INT8 fine-tuning. As expected, the speedups increase with batch size, peaking at a batch size of 8. However, with a batch size of 16, the multi-head attention module becomes another bottleneck, leading to reduced speedup gains.

Table 6: Inference runtimes of one Transformer block in LLAMA3-8B Model, fine-tuned with HALO-2($\overset{\text{H}}{\text{F}}$, $^{\text{H}}\text{E}^{\text{H}}$, $\text{G}^{\text{H}}$) with different batch sizes (BS). We use 512 sequence length.

| BS | BF16 | HALO | Speedup |
|----|------|------|---------|
| 2 | 3.21ms | 2.20ms | 1.46× |
| 4 | 6.19ms | 3.28ms | 1.89× |
| 8 | 13.12ms | 6.88ms | 1.91× |
| 16 | 26.52ms | 14.32ms | 1.85× |

## A.10 Transformer Block Speedups

| | INT8 | | | | FP8 | | | |
|------|------|------|-------|-------|------|------|-------|-------|
| | BS=4 | BS=8 | BS=16 | BS=32 | BS=4 | BS=8 | BS=16 | BS=32 |
| Ideal | 1.62× | 1.69× | 1.75× | 1.70× | 1.32× | 1.37× | 1.35× | 1.28× |
| HALO-0 | 1.52× | 1.63× | 1.67× | 1.63× | **1.27×** | **1.32×** | **1.32×** | **1.24×** |
| HALO-1 | 1.26× | 1.43× | 1.51× | 1.50× | 1.07× | 1.19× | 1.22× | 1.16× |
| HALO-2 | **1.15×** | **1.31×** | **1.36×** | **1.38×** | 0.99× | 1.11× | 1.12× | 1.10× |

Table 7: HALO speedups for different batch sizes (BS) on three consecutive **decoder blocks** of LLAMA3-8B model with 512 sequence length on a single RTX 4090. Ideal shows the speedups when there is no quantization and Hadamard overheads. We **bold** the chosen scheme for each precision.

We evaluate using HALO on three consecutive LLAMA3-8B Transformer blocks (the largest number of blocks that fit on one GPU with batch size 32). Table 7 shows the speedup numbers for different levels in HALO when we apply INT8 and FP8 precisions. Using INT8, the most accurate HALO level, HALO-2, achieves speedups of 1.15× to 1.38× compared to BF16 when using our kernels. The speedup increases with larger batch sizes, as the quantization and Hadamard overheads become less significant, and the matrix multiplications become the primary bottleneck. For FP8, we use HALO-0, which achieves speedups of 1.27× to 1.32×, coming within 5% of the ideal speedup. We note that since FP6 has the same TensorCore peak performance as FP8 (with less read/write overhead), the speedups achieved with FP8 can be considered a lower bound for the potential speedups with FP6 as well.

## A.11 FP8 Linear Layer Speedup

We also benchmark HALO-0 and HALO-1 with FP8 quantization in Table 8. HALO-0 is nearly on par with ideal speedup, peaking at 1.68×. We also provide HALO-1 results when we use with FP6 precision. However, since hardware support for FP6 matrix multiplication is unavailable, we use FP8 matmul instead to provide a lower bound on the FP6 speedup, which peaks at 1.52×.

Table 8: Forward + backward speedups (over BF16) of a linear layer ($4096 \times 4096$) across batch sizes (BS) when we quantize inputs, weights, and output gradients using FP8 representation.

| (RTX-4090) | BS=4 | BS=8 | BS=16 | BS=32 |
|------------|------|------|-------|-------|
| Ideal | 1.20× | 1.65× | 1.70× | 1.78× |
| HALO-1 | 1.07× | 1.47× | 1.48× | 1.52× |
| HALO-0 | 1.12× | 1.62× | 1.63× | 1.68× |

## A.12 HQ-FSDP Speedups

In Table 9 we include detailed speedup numbers for FP8 HALO-0, FP8 HALO-1 and INT8 HALO-2 on four RTX 4090 GPUs, with and without HQ-FSDP.

| | w/o HQ-FSDP | | | | w/ HQ-FSDP | | | |
|---|---|---|---|---|---|---|---|---|
| | BS=4 | BS=8 | BS=16 | BS=32 | BS=4 | BS=8 | BS=16 | BS=32 |
| FP8 Ideal | 1.00× | 1.01× | 1.06× | 1.14× | 1.39× | 1.37× | 1.31× | 1.28× |
| FP8 HALO-0 | 1.00× | 1.01× | 1.05× | 1.12× | 1.39× | 1.37× | 1.30× | 1.26× |
| FP8 HALO-1 | 0.98× | 0.99× | 1.02× | 1.08× | 1.37× | 1.34× | 1.25× | 1.21× |
| INT8 Ideal | 1.00× | 1.02× | 1.14× | 1.43× | 1.40× | 1.40× | 1.48× | 1.65× |
| INT8 HALO-2 | 0.99× | 1.00× | 1.09× | 1.29× | 1.37× | 1.34× | 1.36× | 1.45× |

Table 9: HALO speedups for different batch sizes (BS) on three consecutive decoder blocks of LLAMA3-8B model with 512 sequence length on four RTX 4090 GPUs, with and without HQ-FSDP. `Ideal` shows the speedups when there are no quantization and Hadamard overheads.

### A.13  Qwen Results

To evaluate the effectiveness of HALO on larger models, we extend our GSM8k INT8 experiments to the Qwen-2.5 14B and Qwen-2.5 32B models [44]. Table 10 compares HALO-2 against JetFire [41] and the BF16 baseline. This table shows that HALO-2 outperforms both JetFire and BF16 across both model sizes.

Table 10: Comparison of HALO-2 with JetFire and BF16 on GSM8k for Qwen models.

| Method | Qwen-14B | Qwen-32B |
|---|---|---|
| BF16 | $81.046 \pm 0.5$ | $86.808 \pm 0.3$ |
| JetFire | $81.92 \pm 1.12$ | $87.1 \pm 0.71$ |
| HALO-2 | $\mathbf{82.33 \pm 0.68}$ | $\mathbf{88.40 \pm 0.8}$ |

### A.14  Pre-training Results

We apply HALO to pre-training of TinyLlama-1.1B [47] on the C4 dataset [18]. We select a random Chinchilla-optimal subset [18] (22B tokens), and follow the same hyper-parameters as the original TinyLlama-1.1B [47]. Applying combinations of HALO levels and precisions (INT8, FP4, FP6, and FP8), our results can be summarized as follows:

- FP8 closely matches the base BF16 training even with HALO level 0, both achieving a evaluation loss of $2.55$.
- FP6 converges only with HALO level 2, achieving a final evaluation cross-entropy of $2.70$.
- INT8 and FP4 diverge, no matter the HALO level.

### A.15  MXFP6 Results

In this section, we apply HALO-1 and HALO-2 on the MXFP6 format, comparing them with pure MXFP6 and BF16 training. We consider the LLAMA3-8B model [13] and all three datasets mentioned in the main text. Table 11 shows that HALO-2 closes the accuracy gap with BF16 on the GSM8k [8] dataset.

Table 11: MXFP6 fine-tuning results for LLAMA3-8B[13] on the three datasets, with and without HALO.

| Method | GSM8k | ViGGO | SQL |
|---|---|---|---|
| BF16 | 69.3 ± 0.5 | 94.0 ± 0.3 | 79.9 ± 0.5 |
| MXFP6 | 67.8 ± 0.2 | 93.6 ± 0.6 | 79.6 ± 1.0 |
| MXFP6-HALO1 | 68.0 ± 0.3 | 93.7 ± 0.3 | 80.5 ± 0.2 |
| MXFP6-HALO2 | 68.9 ± 0.5 | 93.5 ± 0.5 | 80.3 ± 0.4 |

