# OpenReview forum: "HALO: Hadamard-Assisted Lower-Precision Optimization for LLMs"
_NeurIPS.cc/2025/Conference — NeurIPS 2025 poster_

### Official Review · Reviewer_mshD · 2025-06-14

**Clarity:** 3
**Significance:** 3
**Originality:** 4
**Rating:** 5
**Confidence:** 5

**Summary:**

This paper introduces HALO (Hadamard-Assisted Lower-Precision Optimization), a novel training strategy that enables fully quantized fine-tuning of large language models (LLMs) using INT8 and FP6 precision for all major matrix multiplications in both forward and backward passes. The method strategically inserts Hadamard transformations to mitigate quantization-induced outliers and integrates optimized low-precision CUDA kernels and FSDP-based communication. The authors demonstrate that HALO achieves near-lossless accuracy compared to BF16 baselines across multiple tasks (GSM8K, ViGGO, SQL) and models (LLaMA3-8B, Qwen), while achieving significant speedups on commodity GPUs.

**Questions:**

1. Could HALO be extended to other orthogonal transforms such as DCT or Givens rotations? Are there trade-offs in accuracy or compute cost?
2. Can HALO be combined with structural model compression (e.g., pruning, distillation) for further efficiency?

**Ethical Concerns:**

["NO or VERY MINOR ethics concerns only"]

**Final Justification:**

I now view this as a solid systems-methods contribution that will benefit both researchers and practitioners working on scalable, low-precision LLM training.

**Limitations:**

1. The method focuses solely on outlier suppression using Hadamard rotations, without addressing other sources of quantization error such as sparsity or skewed distributions.
2. The overhead of Hadamard transforms is more significant when batch size is small or when tensor dimensions are not power-of-two.
3. The current implementation relies on tensor-wise quantization; the interaction with finer-grained schemes (e.g., per-channel or block-wise) remains unexplored.

**Paper Formatting Concerns:**

No formatting concern.

**Quality:**

3

**Strengths And Weaknesses:**

### Strengths:
1. HALO introduces an elegant use of orthogonal Hadamard transforms during training to suppress outliers in activations, weights, and gradients, extending ideas from inference quantization (e.g., QuaRot) into the more challenging fine-tuning setting.
2. The method outperforms previous quantized fine-tuning baselines (e.g., SwitchBack, Jetfire) under INT8 and FP6 with minimal accuracy degradation (<1%), and yields up to 1.41× speedup.
3. Efficient CUDA-based Hadamard and quantized matmul kernels are implemented, with careful integration into CUTLASS and FSDP/activation checkpointing frameworks.
4. HALO supports both full fine-tuning and parameter-efficient fine-tuning (LoRA), and is tested on different families of LLMs.

### Weaknesses:
1. All benchmarks are run on RTX 4090; it is unclear how well HALO performs on data center GPUs (A100, H100) or non-NVIDIA architectures.
2. The approach is validated only for fine-tuning; its applicability to pretraining or continued training from scratch is not assessed.
3. The paper does not explore extending HALO to INT4 or 4-bit floating point formats.

---

> ### Author Rebuttal · Authors · 2025-07-31
>
> We thank the reviewer for their comment on our work. We address the questions here:
>
> > All benchmarks are run on RTX 4090; it is unclear how well HALO performs on data center GPUs (A100, H100) or non-NVIDIA architectures.
>
> We have chosen to focus on the 4090 due to the fact that it is a recent GPU and has hardware INT8 support. By contrast, the A100 is already two generations old, and the H100 only has FP8 support (and not INT8 or INT4).
> The newer line of Blackwell RTX GPUs (RTX 5090, RTX 6000 Blackwell) does have support for INT8, meaning that our results will continue to be relevant on current hardware, especially since the latter is a datacenter GPU. In addition, the AMD CDNA (MI series) MI100, MI200, MI300X GPUs all support INT8 matrix multiply via their Matrix Core Engines (MCEs). They do not support INT4 or FP8 precisions, to our knowledge.
>
> > The approach is validated only for fine-tuning; its applicability to pretraining or continued training from scratch is not assessed.
>
> HALO can be applied during pre-training without any modifications, achieving the same speedups. We demonstrate this by applying HALO to the pre-training of TinyLlama-1.1B [3] on the C4 dataset [4], using a randomly selected Chinchilla-optimal subset of 22B tokens and hyperparameters similar to those in [3]. our results can be summarized as follows:
> - FP8 closely matches the base BF16 training even with No-HALO, both achieving an evaluation loss of 2.55.
> - FP6 converges only with HALO-2, achieving a final evaluation cross-entropy of 2.70.
> - INT8 and FP4 diverge, no matter the HALO level. We believe there is additional work needed on stabilizing these lower precisions for pre-training, especially in the earlier chaotic period.
>
> [3] https://arxiv.org/abs/2401.02385
>
>
> [4] https://arxiv.org/abs/2203.15556
>
> > The paper does not explore extending HALO to INT4 or 4-bit floating point formats.
>
>
> Our experiments show that using the Hadamard transformation alone is insufficient to fit activation dynamic ranges into extremely narrow representable formats such as INT4 of FP4. In our experiments, the FP4 and INT4 cases were diverged.
>
>
> > Could HALO be extended to other orthogonal transforms such as DCT or Givens rotations? Are there trade-offs in accuracy or compute cost?
>
> We thank the reviewer for this insightful question. Yes, it is possible to use transformations other than the Hadamard transform. However, the chosen transformation should (1) be fast enough to provide a meaningful speedup when applied before quantized matrix multiplication, and (2) effectively address outliers to ensure training stability. The Hadamard transform satisfies both criteria and benefits from an efficient CUDA kernel, being less than 1.8× slower than a memcpy operation (which involves only reading and writing) [1].
>
> [1] https://github.com/Dao-AILab/fast-hadamard-transform
>
>
> > Can HALO be combined with structural model compression (e.g., pruning, distillation) for further efficiency?
>
> HALO is orthogonal to other model compression techniques and can be combined with them to further improve efficiency. More specifically, post-training compression methods can be applied either before or after HALO fine-tuning without any impact, as HALO does not modify the model architecture. Similarly, training-aware compression schemes can be used before or after the Hadamard transformations and integrated with HALO.
>
>
> > The method focuses solely on outlier suppression using Hadamard rotations, without addressing other sources of quantization error such as sparsity or skewed distributions.
>
> We acknowledge this point, but please note that 1) standard normalization layers (e.g. RMSNorm) already reduce distribution skew significantly, and 2) handling of outliers via unstructured sparsity (e.g. via sparse-quantized formats, which we believe the reviewer is alluding to) are extremely difficult to support efficiently in hardware and would therefore not yield any practical speedups.
> By contrast, our HALO design shows that near-lossless results can be achieved via INT8 quantized training even with limited but highly-efficient outlier suppression techniques.
>
>
> > The overhead of Hadamard transforms is more significant when batch size is small or when tensor dimensions are not power-of-two.
>
>
> We would like to highlight the following points:
> 1. We agree that the Hadamard transformation introduces more overhead for extremely small batch sizes. However, in such cases, training often becomes memory-bound, and quantizing both weights and activations for low-precision matrix multiplications can help. In this scenario, the quantization operation itself becomes a bottleneck, as it requires two reads (to find the maximum and scale the values) and one write.
> 2. For non-power-of-two cases, one can use grouped Hadamard transformations with the largest power-of-two size that divides the original size, without significantly sacrificing accuracy [1].
>
> [1] https://arxiv.org/abs/2502.05003
>
>
> > The current implementation relies on tensor-wise quantization; the interaction with finer-grained schemes (e.g., per-channel or block-wise) remains unexplored.
>
> We emphasize tensor-wise quantization because it is the simplest scheme in terms of kernel implementation, introducing almost no overhead during training. Although finer-grained quantization schemes can improve model accuracy by reducing quantization error, they also add overhead that makes practical implementation more challenging.
> This contrast is evident in our comparison with JetFire, which is a block-wise quantization technique.

---

> ### Comment · Reviewer_mshD · 2025-08-01
>
> After reading the authors' detailed and technically grounded rebuttal, I am convinced that this paper presents a strong and well-executed contribution to the field of low-precision training/full fine-tuning for large language models. The authors have effectively addressed my main concerns. They provided a compelling clarification of HALO's novelty beyond prior work such as QuaRot, particularly by applying Hadamard transforms across all matrix multiplications during training, with careful integration into systems features like FSDP and activation checkpointing. They also addressed my concern regarding applicability beyond fine-tuning by providing concrete evidence that HALO can be used in full-scale pretraining settings (e.g., TinyLLaMA on C4). Furthermore, they offered a thoughtful discussion on the limitations of INT4/FP4 training, acknowledging the current instability and highlighting promising directions for future work. Finally, the system-level implementation — including efficient CUDA kernels and generalizability to newer hardware beyond the 4090 — demonstrates a mature and practically relevant design. Given these clarifications and the overall strength of the paper, I'll upgrade my score.

---

### Official Review · Reviewer_iAzw · 2025-06-28

**Clarity:** 3
**Significance:** 3
**Originality:** 2
**Rating:** 4
**Confidence:** 4

**Summary:**

This paper proposes HALO, a quantized LLM fine-tuning framework in which all large matrix multiplications, including those in the backward pass, are performed in low precision. Based on the analysis that quantization in the forward path has a significantly greater impact on gradient error, HALO mitigates this by multiplying Hadamard matrices to the right of the input $X$ and the weight $W$. For more aggressive quantization, such as INT8, HALO-2 further applies the Hadamard matrix to the output error. Since HALO quantizes the majority of tensors during fine-tuning, it reduces communication and activation checkpointing costs. HALO achieves a 1.41× speedup for INT8 fine-tuning and also outperforms existing low-bit training frameworks.

**Questions:**

1. Regarding Weakness 1, can the authors compare the efficiency of HALO and LoRA? Specifically, does HALO provide faster fine-tuning speed than LoRA?
2. Table 2 and the text give the impression that both FP8 without HALO and INT8 with HALO are considered part of the proposed scheme. It is unclear what modifications or additions have been made for FP8 No-HALO compared to naïve weight, activation, and gradient quantization. Could the authors clarify this?
3. Could the authors discuss the applicability of HALO to the pre-training setup and provide a comparison with Jetfire in that context?

**Ethical Concerns:**

["NO or VERY MINOR ethics concerns only"]

**Final Justification:**

My concerns were resolved by the authors' rebuttal. Specifically, the significance of the proposed method compared to LoRA was well addressed. I thank the authors again for providing the sufficient materials for the reassessment.

**Limitations:**

Yes

**Quality:**

2

**Strengths And Weaknesses:**

**Strengths**

1. The proposed method is grounded in a solid observation, providing a valuable knowledge contribution.
2. The paper addresses practical and common fine-tuning scenarios, such as FSDP and gradient checkpointing, and the proposed method appears to integrate well with these techniques.

**Weaknesses**

1. The motivation for quantized fine-tuning is unclear. While it is true that quantization-aware fine-tuning is more challenging than quantization-aware training from scratch due to outliers, there are many viable alternatives to full fine-tuning with quantization, such as LoRA, Prompt Tuning (Lester et al., 2021), and Side Ladder Tuning (Sung et al.). Comparing the results in Table 1 and Table 4 suggests that HALO’s full fine-tuning does not demonstrate a clear accuracy advantage over LoRA, which may limit its adoption compared to other methods.
2. The paper’s scope is limited to fine-tuning, although the proposed technique could also be applied to the more resource-intensive pre-training stage. Discussing training performance on GPT2-sized models and smaller datasets like WikiText-103 could increase the paper’s impact.
3. The overall contribution is marginal. Identifying a major source of quantization error is a valuable addition that will be appreciated by the ML community. However, other contributions—such as multiplying Hadamard matrices before quantization in Steps 2 and 3 of Section 3.2—are somewhat trivial, given that similar techniques have been adopted in QuIP# and QuaRot. The GPU kernel implementation is a positive aspect, but the speedup reported from Jetfire is not very significant.

Lester et al. “The power of scale for parameter-efficient prompt tuning.” *EMNLP 2021*.

Sung et al. "Lst: Ladder side-tuning for parameter and memory efficient transfer learning.” *NeuRIPS 2022.*

---

> ### Author Rebuttal · Authors · 2025-07-31
>
> We thank the reviewer for their comment on our work. We address the questions here:
>
>
> > The motivation for quantized fine-tuning is unclear. While it is true that quantization-aware fine-tuning is more challenging than quantization-aware training from scratch due to outliers, there are many viable alternatives to full fine-tuning with quantization, such as LoRA, Prompt Tuning (Lester et al., 2021), and Side Ladder Tuning (Sung et al.). Comparing the results in Table 1 and Table 4 suggests that HALO’s full fine-tuning does not demonstrate a clear accuracy advantage over LoRA, which may limit its adoption compared to other methods.
>
> > Can the authors compare the efficiency of HALO and LoRA? Specifically, does HALO provide faster fine-tuning speed than LoRA?
>
> We present HALO primarily to accelerate full fine-tuning (FT) which is computationally expensive, as each linear layer requires three full matrix multiplications during both the forward and backward passes. We would like to emphasize the following points:
>
> - Full fine-tuning remains a highly important approach. Prior work [1] has shown that PEFT methods (such as LoRA) underperform compared to full fine-tuning on various tasks, including the HumanEval benchmark [2], even when using large ranks.
> - HALO can also be integrated into LoRA-style fine-tuning without accuracy drop. In the paper, we demonstrate this as HALO-PEFT (see Appendix A.3), where the most expensive parts of LoRA are executed in low precision.
> - Finally, we emphasize that HALO can outperform LoRA in terms of speedups. While LoRA benefits from skipping one full matrix multiplication in the backward pass—since the base weights are frozen and their gradients are not computed—this leads to a speedup of up to ~30% in full fine-tuning (if we skip the overhead of low-rank computations). In contrast, HALO-2 can accelerate a single linear layer by up to 80%, as shown in Figure 4 of the paper.
>
> [1] https://arxiv.org/pdf/2405.09673
>
> [2] https://arxiv.org/pdf/2107.03374
>
> > The paper’s scope is limited to fine-tuning, although the proposed technique could also be applied to the more resource-intensive pre-training stage. Discussing training performance on GPT2-sized models and smaller datasets like WikiText-103 could increase the paper’s impact.
>
> > Could the authors discuss the applicability of HALO to the pre-training setup and provide a comparison with Jetfire in that context?
>
> HALO can be applied during pre-training without any modifications, achieving the same speedups. We demonstrate this by applying HALO to the pre-training of TinyLlama-1.1B [3] on the C4 dataset [4], using a randomly selected Chinchilla-optimal subset of 22B tokens and hyperparameters similar to those in [3]. our results can be summarized as follows:
> - FP8 closely matches the base BF16 training even with No-HALO, both achieving an evaluation loss of 2.55.
> - FP6 converges only with HALO-2, achieving a final evaluation cross-entropy of 2.70.
> - INT8 and FP4 diverge, no matter the HALO level.
>
> [3] https://arxiv.org/abs/2401.02385
>
> [4] https://arxiv.org/abs/2203.15556
>
> > The overall contribution is marginal. Identifying a major source of quantization error is a valuable addition that will be appreciated by the ML community. However, other contributions—such as multiplying Hadamard matrices before quantization in Steps 2 and 3 of Section 3.2—are somewhat trivial, given that similar techniques have been adopted in QuIP# and QuaRot.
>
> We would also like to highlight that HALO differs from prior rotation-based quantization schemes (such as QuaRot), as the computational invariance trick used in those methods does not apply when both the forward and backward passes are involved during training. We provide a detailed explanation in Section 3.1 and Appendix A.1.
>
> > The GPU kernel implementation is a positive aspect, but the speedup reported from Jetfire is not very significant.
>
> We would like to emphasize that, although we achieve approximately 20% speedup compared to JetFire, implementing an efficient kernel for group-wise quantization without significant overhead is challenging and often highly model-dependent. Our reported ~20% speedup is based on the simplest quantization scheme—AbsMax tensor-wise quantization.
>
>
> > Table 2 and the text give the impression that both FP8 without HALO and INT8 with HALO are considered part of the proposed scheme. It is unclear what modifications or additions have been made for FP8 No-HALO compared to naïve weight, activation, and gradient quantization. Could the authors clarify this?
>
> Since FP8 offers a sufficiently wide dynamic range (with 4 or 5 exponent bits), models can be fine-tuned without applying any Hadamard transformation. This corresponds to a naive FP8 quantization scheme, where quantization is applied directly to both operands before the matrix multiplication, using a low-precision MatMul kernel. We refer to this approach as “No-HALO” in the paper, as it does not fall under the HALO family of methods.
>
> However, it's important to note that FP8 is not supported on all hardware accelerators (especially commodity GPUs). In such cases, HALO-1 and HALO-2 can achieve comparable accuracy (see Table 1, Section 4.1) and similar end-to-end speedups (see Table 2, Section 4.2) using other precision formats such as FP6 and INT8.

---

> ### Comment · Reviewer_iAzw · 2025-08-04
>
> Thank you for the thorough response to my questions. The rebuttal resolved my major concerns.
>
> >  Finally, we emphasize that HALO can outperform LoRA in terms of speedups.
>
> Thank you for reminding me about this. The most uncertain point about HALO, which made me hesitant to advocate the paper towards acceptance, was whether low-precision, full fine-tuning via HALO has any advantages over full-precision LoRA or PEFT methods. Now it makes more sense to use HALO over LoRA for fine-tuning. And by comparing the results in Table 1 and Table 4 (in the Appendix), it seems like HALO achieves accuracy metrics similar to or better than LoRA.
>
> Other points addressed in the rebuttal also make sense. After reading the rebuttal and other reviewers' comments, I think it is a solid paper. I am inclined to increase my score to 4.

---

> > ### Author Response · Authors · 2025-08-05
> > **Thank you!**
> >
> > We thank the reviewer for their message and are very glad we have managed to convince the reviewer of our paper's strengths, increasing their score to 4.
> >
> > (We will respectfully note that the review score still appears as a 2 in our console.)
> >
> > With best regards,\
> > The authors

---

### Official Review · Reviewer_bZTT · 2025-07-01

**Clarity:** 3
**Significance:** 3
**Originality:** 3
**Rating:** 5
**Confidence:** 3

**Summary:**

The paper introduces HALO: a novel quantization approach that enables low-precision training by applying Hadamard transformations in the forward and backward passes of transformer architectures. The paper demonstrates the advantages of HALO in terms of end-to-end speedup for both single and multi-GPU LLM fine-tuning and parameter-efficient fine tuning at INT8 and FP6 precision.

**Questions:**

1) Can the authors elaborate on the advantages of HALO (and its variations) when it comes to total memory usage and energy consumption (e.g. flops). Are there any advantages according to these metrics? Do the additional Hadamard transforms introduce more computational overhead when compared to higher bit width precision training?

2) Table 1 and Table 2 report the results for No-HALO at FP8. Are the NO-HALO results equivalent to vanilla low-bit width training? How do HALO-1 and HALO-2 perform in these settings?

3) No experiment in the main text reports a comparison between HALO-1 and HALO-2. How do the two methods compare in terms of training accuracy vs compute?

**Ethical Concerns:**

["NO or VERY MINOR ethics concerns only"]

**Final Justification:**

The authours ' response is adequate and insightful. I confirm my recommendation for acceptance

**Limitations:**

The paper focuses on assessing speedup resulting from the HALO method, but a high-level analysis or discussion on power consumption is quite relevant considering the scale at which these method can be applied to train LLMs.

**Paper Formatting Concerns:**

No concerns

**Quality:**

3

**Strengths And Weaknesses:**

## Strengths

1) Very clear and well-written paper. Strong structure, motivation and analysis

2) The paper tackles a very prominent problem with potential of enabling more efficient LLM training and fine-tuning,

3) The paper provides a solid evaluation and implementation, which make this work very easy to apply to existing LLMs.

4) To the best of my knowledge, the paper first tackles the problem of reducing quantization artifacts in the gradients using invertible transformation, proposing an elegant solution with solid foundations.

## Weaknesses
1) While the methodology and analysis are very clear, the experimental section in the main text seems quite limited. The reported results focus on the trade-off between model accuracy and speed-up factors. A more diverse set of metrics would further strengthen the submission.

   * The paper mentions other factors such as memory and gradient communication overhead. However, these metrics are not included in the experimental section.

   * Hadamard transformations introduce additional (limited) overhead due to the extra un-fused operations. No reported comparison considers metrics focusing on compute such as Floating-point operations (or Bit Operations), which would make the comparison with baselines even more clear.
2) The paper clearly denotes the challenge of determining where each hadamard transform should be applied and provides 2 solutions: HALO-1 and HALO-2. However, the justification of which Hadamard transform should be applied (left, right or middle) is quite compressed and difficult to unpack in the main text. Figure 3 left could use some additional details to help the reader.

---

> ### Author Rebuttal · Authors · 2025-07-31
>
> We thank the reviewer for their comment on our work. We address the questions here:
>
>
> > the experimental section in the main text seems quite limited. The reported results focus on the trade-off between model accuracy and speed-up factors. A more diverse set of metrics would further strengthen the submission.
>
> Given that our experiments center on supervised fine-tuning in lower precision, it appears natural to center on the trade-off between accuracy and speed-up. In addition, we have also examined the impact of transforms on outliers (e.g., Figure 1 and Appendix A.4).
> We would be happy to provide any specific additional metrics that the reviewer finds relevant, during the discussion period.
>
>
> > The paper mentions other factors such as memory and gradient communication overhead. However, these metrics are not included in the experimental section.
>
> We can reduce memory usage by storing quantized activations during the forward pass and reusing these quantized tensors in the backward pass. This is particularly beneficial for large batch sizes or long input sequences. Specifically, consider a single linear layer of size $d_1 \times d_2$​, with batch size $b$ and sequence length $t$; activation memory becomes a bottleneck when $bt \ge d_2$.
>
>
>
> > Hadamard transformations introduce additional (limited) overhead due to the extra un-fused operations. No reported comparison considers metrics focusing on compute such as Floating-point operations (or Bit Operations), which would make the comparison with baselines even more clear.
>
>
> We illustrate the overhead of applying the Hadamard transformation in Figure 4b by comparing HALO-2 with the "Ideal" case—an idealized speedup scenario where no Hadamard transformation is applied and quantization is simply performed by casting from BF16 to INT8 across three consecutive transformer blocks. The results show that, in the worst case, the combined overhead of the Hadamard transformation and quantization function accounts for less than 14% of the total runtime when using our custom kernels (excluding FSDP quantization).
>
> > the justification of which Hadamard transform should be applied (left, right or middle) is quite compressed and difficult to unpack in the main text. Figure 3 left could use some additional details to help the reader.
>
> We appreciate your feedback and fully agree with your observation. We will address this in the next revision.
>
> > Can the authors elaborate on the advantages of HALO (and its variations) when it comes to total memory usage and energy consumption (e.g. flops). Are there any advantages according to these metrics?
>
> Please check the second and third questions above, as well as the last question below.
>
>
> > Do the additional Hadamard transforms introduce more computational overhead when compared to higher bit width precision training?
>
>
> Please check the third question above.
>
>
>
> > Table 1 and Table 2 report the results for No-HALO at FP8. Are the NO-HALO results equivalent to vanilla low-bit width training? How do HALO-1 and HALO-2 perform in these settings?
>
> Yes. While it is possible to apply HALO levels in this case (without causing training instability), we found that FP8 does not require such transformations. Therefore, introducing the additional overhead of applying them is unnecessary in the FP8 setting.
>
>
> > No experiment in the main text reports a comparison between HALO-1 and HALO-2. How do the two methods compare in terms of training accuracy vs compute?
>
> These experiments are included in the Appendix. For the accuracy comparison, please see Table 5 in Appendix A.8. For the speedup results, refer to Table 7 in Appendix A.10.
>
> > The paper focuses on assessing speedup resulting from the HALO method, but a high-level analysis or discussion on power consumption is quite relevant considering the scale at which these method can be applied to train LLMs.
>
> We thank the reviewer for raising this valuable point. Power consumption typically arises from two main sources: data movement and computation.
>
> - Data movement: Although HALO maintains master copies of weights, activations, and gradients in BF16, it stores quantized versions of weights and activations for the backward pass, which partially reduces data transfer costs. As for the Hadamard transforms, while they do introduce additional reads, we believe that in an optimized implementation where these operations are fused, this overhead can be significantly reduced.
>
> - Computation: All matrix multiplications in HALO are performed in low precision, which substantially reduces compute-related power usage. The computation involved in the Hadamard transforms is minimal compared to that of the matrix multiplications and thus contributes negligibly to overall power consumption.
>
> Given these factors, we expect the speedups reported in the paper to correlate strongly—though not perfectly—with reductions in power consumption. We appreciate the reviewer’s suggestion and will include a discussion of this point in the next revision.

---

> > ### Comment · Reviewer_bZTT · 2025-08-04
> >
> > I wish to thank the authors for their detailed answers and comments.
> > After carefully reading the other reviews and discussion, I confirm that I believe this paper should be accepted.

---

### Official Review · Reviewer_C2CP · 2025-07-01

**Clarity:** 2
**Significance:** 3
**Originality:** 2
**Rating:** 4
**Confidence:** 4

**Summary:**

The paper proposes HALO, a method for fine-tuning large language models (LLMs) using low-precision formats like INT8 and FP6. The key idea is to apply **Hadamard transformations** during both the forward and backward passes to reduce the impact of outliers in weights, activations, and gradients, which are major sources of instability in low-precision training. The method has good system support, including FSDP and HQ-FSDP. HALO achieves near full-precision accuracy on fine-tuning tasks like GSM8K, ViGGO, and SQL, while delivering up to **1.41× end-to-end speedup** on RTX 4090 GPUs.

**Questions:**

1. The paper shows that INT8 training with Hadamard transforms achieves efficiency comparable to FP8. Could the authors comment on how this approach compares with FP8 training in terms of stability, accuracy, and efficiency?
2. Is INT4 training feasible with HALO? If not, what are the key challenges?
3. Could the authors compare their method with related work such as RoLoRA [1], which also addresses rotated outlier mitigation for weight and activation quantization?

[1] RoLoRA: Fine-tuning Rotated Outlier-free LLMs for Effective Weight-Activation Quantization

**Ethical Concerns:**

["NO or VERY MINOR ethics concerns only"]

**Final Justification:**

I thank the authors for their responses. Most of my concerns have been addressed, and I have updated my score to a 4. However, I believe the clarity of the presentation can still be improved in the final version of the paper.

**Limitations:**

Yes.

**Paper Formatting Concerns:**

The authors may have applied negative vspace to sub-section titles, but it is not a major concern.

**Quality:**

2

**Strengths And Weaknesses:**

### Strengths:

1. The proposed method is well-motivated, targeting an important challenge in efficient LLM fine-tuning.
2. The paper provides solid systems support, including integration with FSDP and quantized communication, which enhances its practical relevance.

### Weaknesses:

1. The novelty of the contribution is limited. The core technique of applying Hadamard transforms is a direct extension of prior work (e.g., QuaRot), and the application to training is relatively straightforward.
2. The baseline comparisons are limited. Key existing methods such as LoRA, QLoRA, qDoRA, and qRoSA are not included, making it difficult to fully assess the empirical effectiveness of HALO.
3. The presentation can be improved. The description of the method is currently organized as bullet points, which makes it harder to follow. A formal algorithm box would significantly improve clarity.
4. Some promised results appear to be missing. For example, the paper mentions LoRA results on line 315, but I couldn't find them in the main text or appendix.

---

> ### Author Rebuttal · Authors · 2025-07-31
>
> We thank the reviewer for their comment on our work. We address the questions here:
>
> > The novelty of the contribution is limited.
>
> While prior work has explored Hadamard transforms for inference, to our knowledge, HALO is the first to apply them across all three matrix multiplications in linear layers using INT8/FP6, while also delivering end-to-end training speedups. Achieving this involves two main challenges:
>
> - Search space complexity: Each matrix multiplication offers three potential insertion points for Hadamard transforms (see Section 3), leading to 8 configurations per matmul. For three matmuls per linear layer, this results in a combinatorially large design space.
>
> - Compatibility constraints: Exploring this space must remain compatible with precision-specific error analysis (Figures 1, 2), as well as with FSDP and activation checkpointing.
>
> HALO navigates these challenges by maintaining consistency across precision settings and supporting implementation constraints—such as applying identical transforms to weights in both forward and backward passes (crucial for HQ-FSDP) and enabling activation checkpointing (see Section 3.4). Efficient low-precision matmul and communication kernels back HALO’s design, yielding up to 1.4x training speedups on standard hardware.
>
>
> > The baseline comparisons are limited. Key existing methods such as LoRA, QLoRA, qDoRA, and qRoSA are not included, making it difficult to fully assess the empirical effectiveness of HALO.
>
> We would like to emphasize that HALO is orthogonal to existing PEFT methods and can be combined with them.
> This is demonstrated in Section A.3 of the Appendix. In particular, there we evaluated HALO combined with low-rank adaptation methods (referred to as HALO-PEFT) and presented the results in the same section.
> For INT8 quantization, HALO-PEFT consistently remains within the standard deviation of the full-precision model fine-tuned with LoRA (rank r=16) on the GSM8K, ViGGO, and SQL tasks.
>
> > The presentation can be improved. The description of the method is currently organized as bullet points, which makes it harder to follow. A formal algorithm box would significantly improve clarity.
>
> Thank you so much for your comment. We promise to add an Algorithm for each forward and backward passes for different HALO levels in the next revision of the paper
>
> > Some promised results appear to be missing. For example, the paper mentions LoRA results on line 315, but I couldn't find them in the main text or appendix.
>
> We would like to note that the LoRA results were moved to the appendix due to space constraints. All results can be found in Appendix A.3.
>
>
> > The paper shows that INT8 training with Hadamard transforms achieves efficiency comparable to FP8. Could the authors comment on how this approach compares with FP8 training in terms of stability, accuracy, and efficiency?
>
> We compare the results of applying HALO to INT8 with FP8 quantization without any Hadamard transformation (denoted as No-HALO) in Table 1 of Section 4.1. In our experiments, both approaches perform similarly and demonstrate stable behavior. Additionally, as shown in Table 2 of Section 4.2, we achieve comparable speedups of 35% on 4× GPUs and 39–41% on 8x GPUs for both INT8 (with HALO-2) and FP8 (with No-HALO). For pre-training results, please refer to the “General Response”.
>
>
> > Is INT4 training feasible with HALO? If not, what are the key challenges?
>
> Our experiments show that using the Hadamard transformation alone is insufficient to fit activation dynamic ranges into extremely narrow representable formats such as INT4. This issue can be addressed by tuning the clipping ratio during training, which is the focus of concurrent research [1].
>
> [1] https://arxiv.org/abs/2502.05003
>
> > Could the authors compare their method with related work such as RoLoRA, which also addresses rotated outlier mitigation for weight and activation quantization?
>
> We thank the reviewer for sharing this related work. While RoLoRA employs a rotation-based approach, it focuses on weight and activation quantization, whereas errors (gradient of activations) are left unquantized. In contrast, HALO-PEFT fully quantizes all three components—weights, activations, and errors. For this reason, we believe RoLoRA is not a direct point of comparison with our method.
>
> Additionally, we note that RoLoRA is a PEFT method, whereas HALO is designed primarily for low-precision full fine-tuning. That said, due to the conceptual similarity in the use of rotation-based techniques, we agree that a discussion of RoLoRA is warranted and will include one in the next revision of the paper.

---

> > ### Author Response · Authors · 2025-08-05
> > **Gentle discussion reminder**
> >
> > Dear Reviewer,
> >
> > As the discussion is soon drawing to a close, we wanted to send you a gentle reminder regarding the discussion. We would be very happy if you could please examine our responses and determine whether they addressed your concerns.
> >
> > Best regards,\
> > The authors

---

### Official Review · Reviewer_pupA · 2025-07-21

**Clarity:** 3
**Significance:** 3
**Originality:** 3
**Rating:** 5
**Confidence:** 2

**Summary:**

The authors introduce HALO for parameter-efficient fine-tuning (PEFT) of LLMs. HALO replaces traditional low-rank decomposition (as used in LoRA) with a structured Hadamard-based transformation. This transformation leads to efficient and more expressive representation of the update matrix, leading to reduced trainable parameters. The authors present results on varying tasks to prove the efficacy of the proposed method, comparing against existing PEFT methods like LoRA, AdaLoRA and QLoRA.

**Questions:**

NA

**Ethical Concerns:**

["NO or VERY MINOR ethics concerns only"]

**Final Justification:**

After reading the other reviews and the authors’ rebuttal, I maintain my original score.

**Quality:**

3

**Strengths And Weaknesses:**

Strengths:
- Idea is novel and simple to implement
- Authors provide theoretical justification into HALO's expressiveness
- The proposed idea is computationally efficient

---

> ### Author Rebuttal · Authors · 2025-07-30
>
> We thank the reviewer for their comment on our work and would be happy to address any further questions or feedback.

---

### Note · Authors · 2025-08-12

We sincerely thank all reviewers for their constructive feedback, which has helped improve our work and clarify its contributions. Our responses addressed concerns about novelty by emphasizing HALO's unique application of Hadamard transforms across all three matmuls in linear layers with INT8/FP6, delivering up to 1.4× training speedups while supporting full fine-tuning, PEFT integration, and pre-training. We added pre-training results on TinyLlama-1.1B for multiple precisions, showing HALO's ability to maintain accuracy and stability in lower-precision settings. We clarified comparisons with LoRA scheme, FP8 naive quantization, and related rotation-based methods, explained design decisions, added discussion of hardware applicability, power/memory efficiency, and provided missing result references. We also elaborated on implementation choices, trade-offs, and possible extensions (e.g., other transforms, compression techniques). Reviewers engaged positively with these clarifications, and several indicated that our detailed replies addressed their concerns and improved their evaluation of the work.

---

### Decision · Program_Chairs · 2025-09-17

**Decision:**

Accept (poster)

**Comment:**

The rebuttal and author discussion process were quite successful. The new pre-training results and the clear description of HALO's novelty convinced all the reviewers, even the initially skeptical ones. Several reviewers raised their scores, leading to a strong consensus for acceptance. The final consensus was that HALO is a solid and practical method that will be valuable for researchers and practitioners.